# Hedgehog Signaling in Gonadal Development and Function

**DOI:** 10.3390/cells12030358

**Published:** 2023-01-18

**Authors:** Iman Dilower, Asef J. Niloy, Vishnu Kumar, Ayushi Kothari, Eun Bee Lee, M. A. Karim Rumi

**Affiliations:** Department of Pathology and Laboratory Medicine, University of Kansas Medical Center, Kansas City, KS 66160, USA

**Keywords:** hedgehog, ovary, granulosa cells, theca cells, steroidogenesis, follicle development

## Abstract

Three distinct hedgehog (HH) molecules, (sonic, desert, and indian), two HH receptors (PTCH1 and PTCH2), a membrane bound activator (SMO), and downstream three transcription factors (GLI1, GLI2, and GLI3) are the major components of the HH signaling. These signaling molecules were initially identified in *Drosophila melanogaster*. Later, it has been found that the HH system is highly conserved across species and essential for organogenesis. HH signaling pathways play key roles in the development of the brain, face, skeleton, musculature, lungs, and gastrointestinal tract. While the sonic HH (SHH) pathway plays a major role in the development of the central nervous system, the desert HH (DHH) regulates the development of the gonads, and the indian HH (IHH) acts on the development of bones and joints. There are also overlapping roles among the HH molecules. In addition to the developmental role of HH signaling in embryonic life, the pathways possess vital physiological roles in testes and ovaries during adult life. Disruption of DHH and/or IHH signaling results in ineffective gonadal steroidogenesis and gametogenesis. While DHH regulates the male gonadal functions, ovarian functions are regulated by both DHH and IHH. This review article focuses on the roles of HH signaling in gonadal development and reproductive functions with an emphasis on ovarian functions. We have acknowledged the original research work that initially reported the findings and discussed the subsequent studies that have further analyzed the role of HH signaling in testes and ovaries.

## 1. The Hedgehog System

The hedgehog (HH) signaling was identified for their roles in the body segmentation of *Drosophila melanogaster* [1]. The name HH represents *Drosophila* larvae’s spiked phenotype in the cuticle, which appears like the HH animal’s spikes [2,3]. Subsequent studies found that the HH signaling is highly conserved across species and is essential for organogenesis [4]. The HH system is composed of three HH ligands (sonic HH [SHH], desert HH [DHH]), and indian HH [IHH]), two HH receptors (PTCH1 and PTCH2), a transmembrane activator protein smoothened (SMO), and three downstream glioma-associated oncogene homologue (GLI1, GLI2, and GLI3) transcription factors (TFs). HH signaling activates GLI TFs, which translocate into the nucleus, and initiate the transcription of the target genes [5]. As the HH ligands have common receptors, and common downstream signaling molecules, some overlapping functions among HH molecules have been detected in many organs and tissues. However, individual HH also carryout distinct physiological roles [2,6].

The major role of HH signaling is limited to embryonic development, particularly in organogenesis. SHH plays an essential role in neuronal development, DHH acts on the gonadal development and steroidogenesis, and IHH regulates various developmental functions including bone development, as well as ovarian steroidogenesis and folliculogenesis [7,8]. Dysregulation of HH signaling results in the developmental defects in brain, face, and other midline organs [9,10,11,12]. Proper development of skeleton, musculature, gastrointestinal tract, and lungs does not occur in the absence of HH signaling [13,14,15,16,17]. The revival of HH activities in adult life has been detected during tumorigenesis, and inhibition of HHs has been targeted for cancer therapy [18]. Most importantly, studies have demonstrated that HH signaling plays an important role in both prenatal and postnatal gonadal development and function [19,20,21]. HH signaling also regulates the gonadal functions in adult life including steroidogenesis, spermatogenesis, and folliculogenesis [20,21,22,23,24,25]. Loss of HH signaling can lead to infertility in both males and females [26,27,28]. In this article, we have introduced the basic aspects of HH signaling and elaborated the role of HH signaling in gonadal development and function.

## 2. The Hedgehog System in Health and Diseases

### 2.1. Physiological Roles of Hedgehog Signaling

HH signaling is primarily known for its essential roles during embryonic development [2]. It also plays important regulatory roles in adult life, particularly in gonadal functions. Aberrant activation of HH signaling has been detected in cancers [29]. HH signaling molecules are differentially expressed in various tissues resulting in tissue-specific regulatory functions. Different HH molecules may also express in the same tissue and share common downstream signaling molecules; therefore, these signaling systems may exhibit tissue-specific unique functions as well as overlapping and combined functions among different HHs [8,30,31].

SHH is widely expressed in different organs and tissues, and it is the most potent HH ligand among the three [32]. It is known for its role in the developmental patterning of neural tube and limb buds [33,34,35,36,37]. SHH regulates the proliferation and differentiation of neuronal precursors and are involved in the development of cerebral cortex studied in mice [35] (Figure 1). It also plays a critical role in the development of axial structures including the floor plate [38]. It regulates the patterning of limbs as well as the formation of bones [7,16,33]. It is suggested that SHH signaling works with BMP4 to regulate the growth of epithelial stem cells throughout embryonic development [39,40]. Moreover, the targeted knockout of *Shh* gene (*Shh^KO^*) in the adrenal cortex leads to adrenocortical hypoplasia, indicating that SHH is essential for the development of the adrenal gland [41].

DHH is primarily involved in the development of the gonads [19,20,28,42] (Figure 1). In male mice, DHH signaling regulates the development of testes, steroidogenesis, and spermatogenesis [43,44]. *Dhh* gene knockout (*Dhh^KO^*) male mice are infertile due to the absence of mature sperms [43,44]. Expression of DHH is detected in the Sertoli cells (SCs) of testes, and acts on the development and differentiation of peritubular myoid cells (PTMCs) and fetal Leydig cells (LCs) [45]. Defective development of LCs in *Dhh^KO^* male mice is associated with pseudo-hermaphroditism, characterized by incomplete masculinization of the testes and the male genital tract [42]. DHH is also expressed in Schwann cells, the glial cells of peripheral nerves, which play an important role in nerve sheath formation [46]. Disruption of DHH signaling leads to peripheral neuropathy associated with mini-fascicle formation [47].

In contrast to SHH and DHH, IHH plays a predominant role in the development of bones and cartilages [48]. It is expressed in prehypertrophic and hypertrophic chondrocytes of developing endochondral bones and synchronizes chondrogenesis as well as osteogenesis during endochondral ossification [49]. *Ihh* gene knockout (*Ihh^KO^*) mice suffer from a lack of mineralization of bones and fails to form the osteoblasts in endochondral bones, which is required for skeletal growth [7,49]. IHH also play an important role in the formation of synovial as well as temporomandibular joints [50]. While DHH alone plays a key role in testes, the presence of both DHH and IHH are required for the ovarian functions [19] (Figure 1).

HH molecules act alone in several target tissues, whereas they exhibit overlapping roles in others. Therefore, ‘the loss of function’ phenotypes due to the loss of any single HH differ from the combined loss of multiple HH molecules [8,30,31]. The loss of both SHH and IHH results in lack of SMO expression in early embryos associated with lethal defects in cardiac development as well as extraembryonic vasculogenesis [30,31]. On the other hand, both IHH and DHH are expressed in ovarian granulosa cells (GCs) and play important roles in steroidogenesis and follicle development [8]. It has been reported that the loss of both DHH and IHH in GC results in the failure of theca cell (TC) development, defective steroidogenesis, and infertility in female mice [8].

### 2.2. Abnormal Hedgehog Signaling and Developmental Disorders

HH signaling plays a vital role in organogenesis during embryonic development, thus disruption of the HH pathways results in various developmental disorders. In most cases, the downregulation of HH signaling is implicated in birth defects [9,51] (Figure 2).

Mutations in the human *SHH* gene that downregulate *its function* are the common causes of sporadic and inherited holoprosencephaly, characterized by incomplete separation of the left and right cerebral hemispheres [52,53] (Figure 2). In contrast, an increased function of SHH signaling has been associated with exencephaly and spina bifida [54]. Moreover, an aberrant SHH signaling cause ciliopathies, a disorder in ciliary functions [55]. Based on the findings in ciliopathy mouse models, it has been suggested that ciliary dysfunctions in the inner ear due to the loss of SHH signaling can lead to hearing loss [55]. Aberrant HH signaling have been linked to the development of cancers [4,56,57,58] (Figure 2). Many types of solid and hematological cancers are found with the hyperactivation of HH signaling [59]. The SHH signaling is strictly regulated in adult tissues and upregulation of the SHH signaling has been found to be oncogenic [60,61]. An ectopic expression of the SHH alone can induce basal cell carcinoma in mice [62] (Figure 2). Recent studies have focused on the SHH signaling for molecular targeting of cancer therapy [63,64].

DHH plays a crucial role in male germline development in embryos and spermatogenesis in adults [20,42]. Mutations in DHH have been found to be associated with gonadal dysgenesis in males and development of seminoma [65]. In contrast, disruption of IHH signaling has been implicated in defective bone formation as well as abnormal hematopoiesis or angiogenesis [3,44]. Mutations of IHH gene may result in the defective skeletal development such as brachydactyly [66] and acrocapitofemoral dysplasia (short limbs, large head, and narrow thorax) [67] (Figure 2). Hereditary multiple exostoses are another IHH-related growth abnormality, which is characterized by a smaller skeleton with multiple cartilage-capped bony outgrowths as well as benign bone growth (exostoses) in endochondral bones [68]. Recent studies also suggest that loss of IHH signaling may also be associated with defective steroidogenesis in females [8,23,69].

## 3. Hedgehog Expression, Processing, and Signaling Pathways

HH signaling functions in a unique two compartment system: a specific cell population express the HH molecules, and the secreted HH molecules act on target cells that possess the PTCH receptor, SMO activator, and GLI TFs [70]. The active form of HH molecules is secreted from the expressing cells after various post-translational modifications (PTMs), as described in the following sections (Figure 3).

### 3.1. Posttranslational Modifications of Hedgehog Proteins

The HH ligands are translated as ~46 kDa precursor peptides and undergo several post-translational modifications (PTMs) before they are secreted in an active form [71]. Such PTMs are highly conserved and apply to all mammalian HH isoforms [71]. The PTMs of the HH proteins determine the way HH molecules are presented on the target cell surface [71,72]. HH polypeptides are transferred to the endoplasmic reticulum (ER) and Golgi apparatus for autoprocessing [71,72] (Figure 3). The autoprocessing starts with the removal of the signal peptides from HHs, followed by an internal cleavage that generates a ~19-kDa N-terminal and another ~25-kDa C-terminal fragment [73] (Figure 3). The N-terminal fragments are modified by the addition of a cholesterol group at the C-terminus, and serves as the active HH ligands [73]. Cholesterol transferases and the HH acyltransferase, which are in the endoplasmic reticulum (ER), further modify the N-terminal part of HH proteins [73,74,75,76,77,78,79,80] (Figure 3). Remarkably, the only signaling proteins known to be covalently changed by cholesterol moiety are the HHs [73]. A palmitic acid moiety is added to the N-terminus of HH signaling domain by acetyltransferase known as SKI [73]. After the cholesterylation and palmitoylation, the signaling domain is secreted as an active HH molecule. On the other hand, the C-terminal HH fragments are involved in the autoprocessing of HH molecules and undergo rapid degradation [73,77] (Figure 3).

### 3.2. Signal Transductions Mediated by Hedgehog Proteins

One of the distinctive characteristics in vertebrate HH signaling is the relationship between primary cilia and HH signaling [81]. The primary cilium, a specialized organelle protruding from the cell surface, have been critical to the distribution and function of mammalian HH signaling [81,82] (Figure 4).

The major target molecules of HH proteins are PTCH1, PTCH2, SMO, GLI1, GLI2, and GLI3. The initial step is mediated by two transmembrane receptors: one is either PTCH1 or PTCH2, and the other is SMO [83] (Figure 4C). In a canonical pathway, HH ligands bind to PTCH1, which leads to the release of SMO from the PTCH1-mediated inhibition [83]. Free SMO activates the GLI TFs that induce targeted gene regulation [83,84] (Figure 4C). While GLI1 primarily acts as an activator, GLI2 and GLI3 can act either as an activator or as a repressor. In absence of the HH-mediated activation of SMO, GLI2 and GLI3 are phosphorylated by PKA, GSK-3β, and CK1, which leads to the cleavage of these GLI proteins to generate their repressor forms, GLI2R and GLI3R [44] (Figure 4B).

HH signaling can be impacted by several interacting molecules [85]. KIF7 and SUFU can influence the stability and transcription activity of the GLIs [86,87,88]. KIF7 can exert either a positive or a negative regulatory effect on GLI functions, while SUFU acts as an inhibitor of GLIs [89]. In addition, SUMO has been found to modify SMO and GLI family members to stabilize and activate the target proteins. CDON and BOC bind to HH proteins and play a positive role in HH signaling [90,91]. In addition, cell surface protein GAS1 has been shown to positively regulate and HHIP has been shown to negatively regulate the HH signaling [92,93].

In contrast to the canonical HH signaling, the non-canonical HH signaling can be mediated by two distinct mechanisms. The type I non-canonical HH signaling pathway is SMO-independent but GLI dependent [5,94] (Figure 5A,B). Whereas, the type II HH signaling is SMO-dependent but independent of the GLI-mediated transcriptional signaling [5,94,95] (Figure 5C).

## 4. Expression and Regulation of HH System in the Gonads

While DHH regulates the development and functions of male gonads, the development and functions of female gonads are dependent on both DHH and IHH. DHH is expressed in the SCs of developing testes starting from a mid-embryonic stage, whereas DHH and IHH are expressed in the GCs of activated follicles. While SC-derived DHH acts on LCs and spermatocytes, GC-derived DHH and IHH act on TCs.

### 4.1. Expression and Localization of Hedgehog System

Both SCs and the male germ cells are enclosed in the seminiferous tubules by a basement membrane formed by the peritubular myoid cells (PTMCs). SRY induces the expression of SOX9 that differentiates the SCs [96,97]. Differentiated SCs of fetal testes express DHH starting from embryonic day (E)11.5 [23,98] (Figure 6A). HH target molecules are expressed on the LCs, and PTMCs (Figure 6B). HH receptor PTCH2 is highly expressed in spermatocytes and helps to mediate the DHH activity in germ cell development [99].

GCs in dormant primordial follicles (PdFs) do not express the HH molecules; both DHH and IHH are induced in the GCs of activated follicles starting from the primary follicle stage (PrF) stage [19,100] (Figure 6C). However, the downstream targets of HH signaling, including PTCHs, SMO, and GLIs, are located in the TCs, which suggest that GC-derived HHs act on the TCs [19,100] (Figure 6C).

### 4.2. Regulation of Hedgehog Expression

During the development of the gonads, WT1, GATA4, GATA6, SOX9, and SRY are involved in regulating the transcription of *Dhh* in SCs [97,101]. Among these transcriptional regulators, SRY and SOX9 plays the major role in inducing DHH expression development of testes [97,101]. Patients and mouse models carrying inactivating mutations in *Sox9* gene exhibit the sex reversal of the XY chromosome background [102,103]. SRY not only regulates the proliferation, differentiation, and functions of SCs, it also induces the expression of Sox9, which continues the differentiation of the testis [97].

The expression of DHH and IHH remain undetectable in ovarian PdFs and is induced in the GCs of activated follicles. The expression of both HHs remain higher in the GCs of early-stage follicles but downregulated in more developed preovulatory follicles [19,74]. Studies have demonstrated that after the induction of LHCGR signaling, DHH and IHH mRNA levels are decreased to the basal levels and remains low until the ovulation occurs [19,74]. During the preovulatory period, the expression of PTCH1 and GLI1 mRNAs is also reduced significantly in the TC-interstitial compartment [19,74]. Another important aspect of ovarian HH regulation is the role of a bidirectional signaling between GCs and oocytes [73]. Although the components of HH pathways are located in the somatic cells (GCs and TCs) of ovarian follicles, the expression of HH molecules are regulated by the oocyte-derived factor such as GDF9 [73]. It has been reported that the expressions of DHH, IHH, and GLI1 are significantly decreased in the ovaries of Gdf9KO mice that lack oocytes [73]. When GDF9 is added to oocyte-depleted Gdf9KO ovaries, the expression of DHH, IHH, and GLI1 increases, indicating that GDF9 plays a crucial role in the expression of HH ligands in GCs [77]. In a recent study, we observed that the expression of Ihh and Hhip in neonatal rat ovaries is dependent on the estrogen receptor β (ERβ) [21].

## 5. Hedgehog Signaling in Gonadal Development

Sexually dimorphic features are prominent between the male and female gonadal systems [23]. HH signaling plays a decisive role during the development of the gonads, reproductive tracts, and external genitalia [23]. Differential HH functions are important for the sexually dimorphic gonadal development and function [23] (Figure 7). However, the morphogenetic events, including the development of gonad-specific cell types, structure of the reproductive tract, and external genitalia, may need further involvement of the endocrine and paracrine signaling pathways [20,28,98,104].

### 5.1. Dimorphic Development of Male and Female Reproductive Sytem

The sexually dimorphic structures of the male and female reproductive system develop in three distinct steps. In the first step, sex determination occurs when X or Y chromosome-carrying sperm fertilizes an X chromosome-carrying oocyte, the XX gametes develop to females, and the XY gamete develop to males [105]. Following the chromosomal determination, formation of primary or gonadal sex begins and the testis and the ovaries are specified [105] (Figure 7). The sex-determining region of chromosome Y (SRY) is responsible for the morphogenesis of testis [106,107]. In the absence of the SRY expression, XY embryos develop ovaries instead of testes [106,107] (Figure 7).

The final step of sex differentiation occurs with the development of the reproductive tracts and external genitalia. The testes express the anti-Müllerian hormone (AMH) and androgens, which induces Müllerian duct regression and Wolffian duct differentiation into the epididymis, vas deferens, and seminal vesicles in male embryos [106,107] (Figure 7). As the ovaries express very low levels of AMH and androgens, the Wolffian duct regresses while the Müllerian duct persists and becomes the oviduct, uterus, cervix, and upper section of the vagina [108]. While androgens stimulate the development of male external genitalia, the deficiency of androgens in females leads to the development of female genitalia [108] (Figure 7). The role HH signaling in the dimorphic development of male and female gonads and other reproductive organs are discussed in the following sections.

### 5.2. Hedgehog Signaling in Male Gonadal Development

DHH plays a key role in the development of male gonads [20,109]. Development of the PTMCs and FLCs are dependent on the SC-derived DHH [42,110,111] (Figure 8). SC-derived DHH acts on the PTCH receptors expressed on LCs to induce the differentiation synthesis of androgens. During the development of male gonads, SHH is expressed in the Wolffian duct epithelium, and PTCH1 and GLI1 are expressed in the mesonephric mesenchyme, which may also contribute to the process [23] (Figure 8).

In the absence of DHH, the development of PTMCs and FLCs is defective in Dhh^KO^ mice, which leads to the disorganized structure of the corda testes [28,42,109]. Irregularly shaped SCs, abnormal PTMCs, discontinued basal lamina, and germ cells positioned outside the cords are the histological features of Dhh^KO^ mouse testes [42,109]. DHH also regulates the proliferation and differentiation of adult LCs (ALCs) [112]. Dhh^KO^ mouse testes possess the undifferentiated LCs, resulting in testosterone deficiency [28,42]. However, Dhh^KO^ XY mice may develop a variety of testicular phenotypes depending on the genetic background; ultimately, they become infertile due to the lack of mature sperms [20,42,109].

### 5.3. Hedgehog Signaling in Female Gonadal Development

Interactions between the oocytes, GCs, and TCs are essential for the development and maturation of ovarian follicles [19]. HH signaling pathways represent a good example of interactions among the GCs, TCs, and oocytes [19] (Figure 9). TCs develop surrounding the secondary follicles (ScFs), containing two or more layers of GCs [113]. The precursors of TCs arise from two sources [69]. The androgen-producing TCs in the basal lamina are generated from the mesonephros, whereas the remaining TCs surrounding those develop from the ovarian mesenchyme [42] (Figure 9).

During follicle development, the mesenchymal compartment appears to be a predominant target of HH ligands [69]. Activated GCs express DHH and IHH, which are essential for the proliferation and differentiation of TC precursor cells [69]. HH signaling may also play an important role in protecting the ovarian follicle reserve. A recent study has demonstrated that the inhibition of HH signaling with an inhibitor (GANT61) reduced the mouse ovarian PdF count [114]. However, further studies are required to confirm these findings and to clarify the underlying mechanisms.

## 6. Hedgehog Mediated Regulation of Gonadal Functions

### 6.1. Hedgehog Regulation of Testicular functions

DHH regulates spermatogenesis and the maturation of sperms [115]. DHH released from the SCs acts on PTCH1 expressed in PTMCs and FLCs, as well as endothelial cells in fetal testicular interstitium, and activates GLI TFs [20,28,42,104]. DHH mutant rats suffer from defective development of FLCs as well as inadequate production of androgens [45] (Figure 10). As expected, exposure to HH inhibitors slow down the formation of FLCs in fetal gonad explants [28]. Cyclopamine-mediated inhibition of DHH signaling was found to downregulate the expression of PTCH1 associated with the disruption of LC differentiation [28] (Figure 10).

It has been demonstrated that DHH signaling initiates the development and maturation of FLCs in mice by upregulating the expression of SF1 [116,117]. In turn, SF1 increases the expression of the steroidogenic enzymes CYP11A1 and HSD3B1 [116,117]. Dhh^KO^ XY gonads lose the expression of both PTCH1 and CYP11A1 [20,28,118]. In mouse embryonic testes, DHH signaling induces the formation of FLCs from the SF1-positive FLC progenitors [119]. Expression of GLI1 has also been detected in the SCs, implying that DHH signaling may also regulate SCs in an autocrine manner [120]. However, it was found that the inhibition of HH signaling does not impair the differentiation of SCs [118].

### 6.2. Hedgehog Regulation of Ovarian Functions

Ovary-specific either Dhh^KO^ or Ihh^KO^ female mice were found to be fertile and demonstrated the presence of corpora lutea in their ovaries [8]. However, the combined Dhh^KO^ and Ihh^KO^ resulted in defective TC development, follicular arrest at preantral phases, and failure of ovulation [8]. TC differentiation was found to be dependent on GC-derived DHH and IHH (Figure 11). Nevertheless, the nature of the overlapping roles of DHH and IHH in ovarian TC cell differentiation remain unclear [8].

GC-derived DHH and IHH activate GLI TFs, which translocate to the nucleus of activated TC precursor cells. Activated GLIs induce the expression of target genes that mediate the proliferation and differentiation of the TC percussor cells to mature TCs [69] (Figure 11). Mice lacking the expression of both DHH and IHH in GCs (Dhh^KO^; Ihh^KO^) fail to develop the TC cells, and suffer from impaired steroidogenesis and infertility due to failure of ovulation [8]. However, such a phenotype is not observed in either Dhh^KO^ or Ihh^KO^ mice [8]. In the absence of HH signaling, neither α-SMA, HSD3β or CYP17A1 was detected in ovarian follicles indicating that development of both mesenchyme or mesonephros derived TCs is dependent on the GC-derived DHH and IHH [8]. Remarkably, Ihh^KO^ female mice, but not the Dhh^KO^ mice, showed progressively lower levels of dehydroepiandrosterone, testosterone, and progesterone [8]. It was also associated with an altered expression of steroidogenic enzymes, which indicates that IHH plays a crucial role in regulating ovarian steroidogenesis [8].

## 7. Undecided Issues in Hedgehog Functions in the Gonads 

It remains unknown whether the phenotypic consequences on spermatogenesis in Dhh^KO^ mice are the direct effects of DHH deficiency or an indirect effect of androgen deficiency due to abnormal development and functions of LCs. It is also suggested that DHH signaling has distinct roles on the development of LCs, and PTMCs that are independent of androgens. While the male gonadal functions are regulated by DHH alone, steroidogenesis and folliculogenesis in the female gonad are regulated by both DHH and IHH signaling. We observed that while the expression of IHH in GCs is regulated by ERβ, the expression of DHH is independent of the ERβ signaling. We suspect that IHH and DHH may execute a differential regulatory role in ovarian steroidogenesis and/or folliculogenesis. Further studies are required to distinguish between the roles of IHH and DHH in the ovarian follicle development and ovarian functions.

## Figures and Tables

**Figure 1 cells-12-00358-f001:**
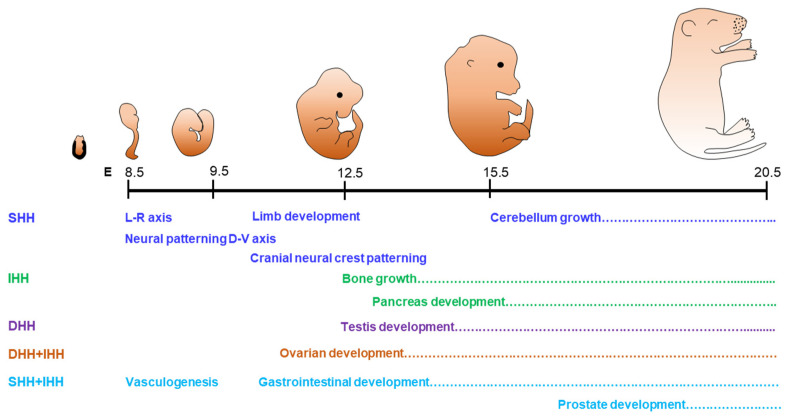
Role of hedgehog signaling in organogenesis. Hedgehog (HH) signaling is highly conserved across species and plays a major role in organogenesis during the embryonic life. The schematic presentation shows the days in mouse embryonic development (considering the day of mating plug positive as E0.5). Name of the functions and the dotted lines indicate the duration in embryonic days when SHH, IHH, and DHH control the developmental processes. Both IHH and DHH play an important role in the development of gonads, while SHH and IHH contributes to vasculogenesis, gastrointestinal development, and the development of prostate gland.

**Figure 2 cells-12-00358-f002:**
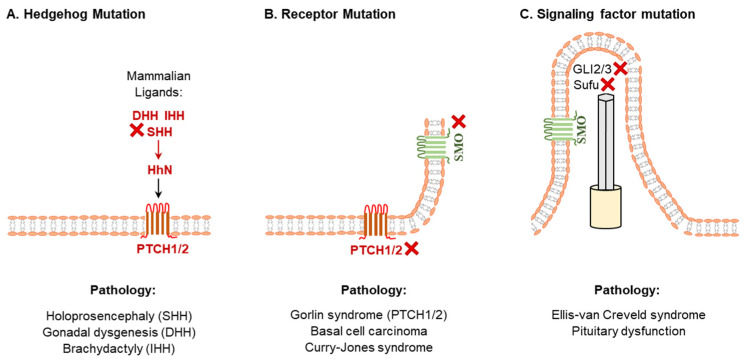
Diseases due to abnormal hedgehog signaling. Mutations in HH coding sequence (**A**), HH receptors (**B**) or mutations in the downstream signaling molecules (**C**) of HH signaling result in various pathological conditions. In the lower panels, a selected group of common pathological conditions are indicated under the corresponding mutations.

**Figure 3 cells-12-00358-f003:**
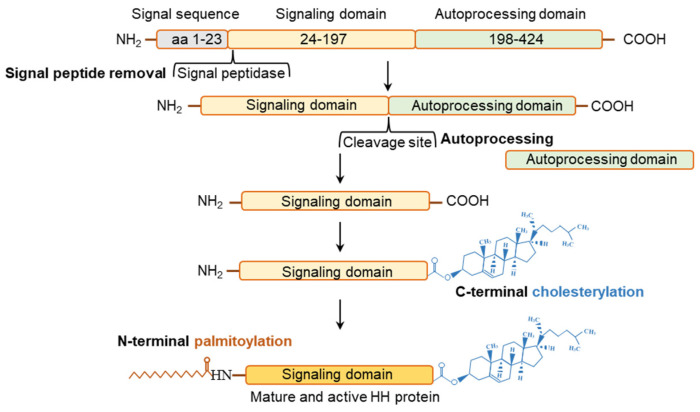
Posttranslational modifications of hedgehog molecules. Posttranslational modification in each of the three HH molecule is similar and highly conserved. The initial HH polypeptide is stepwise cleaved into three domains. After the cleavage of signal sequence (~aa 1–23), the remaining polypeptide is cleaved into a N-terminal signaling domain (~aa 24–197) and a C-terminal autoprocessing domain (~aa 198–424). The Signaling domain undergoes C-terminus cholesterylation and N-terminal palmitoylation before it is secreted in its mature and active form. The autoprocessing domain catalyses the intramolecular chlesterol tranfer reaction necessary for choesterylation of the signaling domain.

**Figure 4 cells-12-00358-f004:**
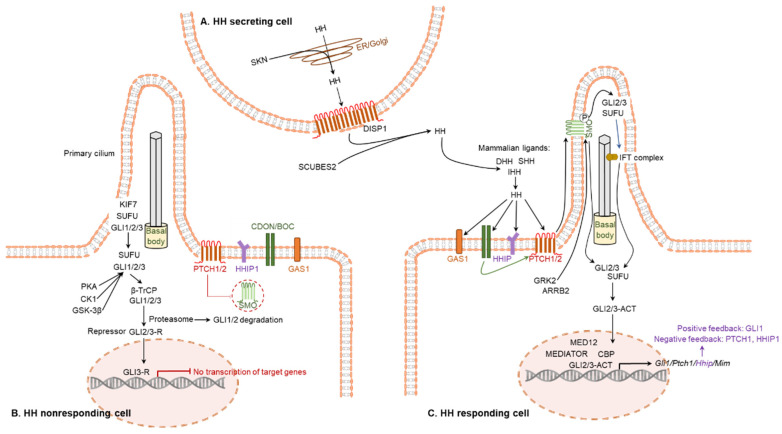
Canonical hedgehog signaling pathway. (**A**) The active form of HH molecules are secreted from the HH expressing cells. (**B**) In the absence of HH ligand, PTCH inhibits SMO. This is associated with GSK-3β, CK1, and PKA-mediated phosphorylation of GLI, which forms a truncated form of GLI repressor. The repressor GLI translocates to the nucleus in order to inhibit the transcription. (**C**) Active form of HH ligand binds to PTCH receptor on the responding cells and the HH ligand-dependent interaction with PTCH and SMO results in release of SMO activator. SMO controls the processing of GLI factors, activate GLI, and initiate the cascade of downstream signaling pathways. Activated GLI translocates to the nucleus and initiates the transcription of HH target genes.

**Figure 5 cells-12-00358-f005:**
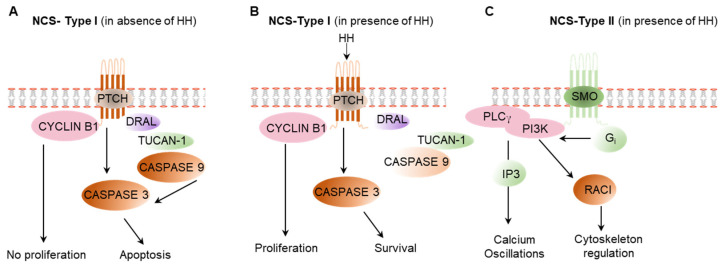
Non-canonical hedgehog signaling pathways. (**A**) In the absence of HH ligand, PTCH forms a complex with CYCLIN B1, DRAL, TUCAN-1, and CASPASE 9 that activates CASPASE 3, which induces apoptosis. In the absence of HH ligand, CYCLIN B1 remain inactive in the proteome complex and do not induce cell proliferation. (**B**) In presence of HH ligand, PTCH breaks the interaction with CYCLIN B1, DRAL, and CASPASE 9 and CASPASE 3, which activate the cell proliferation. CASPASE 3 is not be activated, which results in cell survival. (**C**) HH-activated SMO causes the dissociation of Gi, which activate PI3 kinase, which results in the regulation of cytoskeleton and calcium signaling. NCS: non cannonical signaling of HH.

**Figure 6 cells-12-00358-f006:**
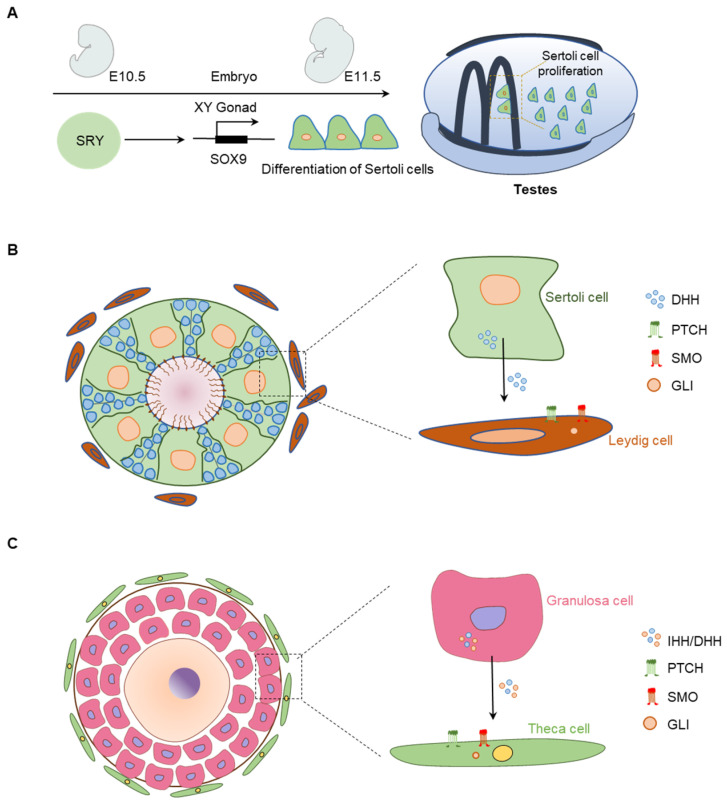
Expression and localization of HH signaling molecules in the gonads. (**A**) Role of the DHH signaling in fetal testis’ morphogenesis and differentiation. Schematic presentation of embryonic day 10.5 (E10.5) to E11.5 fetal mouse testis depicting SRY-induced SOX9 expression. SRY and SOX9-induced factors mediate proliferation and differentiation of Sertoli cells (SCs), which express DHH. (**B**) DHH secreted from the SCs acts on the PTCH and SMO expressed in Leydig cells (LCs) and induce activation of GLI transcription factors. DHH signaling induce steroidogenesis in differentiated LCs. (**C**) In the ovary, the granulosa cells express IHH/DHH, which act on PTCH-positive TC precursor cells. HH binding to PTCH1 releases SMO, which activate the GLI2 TFs and differentiate the TCs.

**Figure 7 cells-12-00358-f007:**
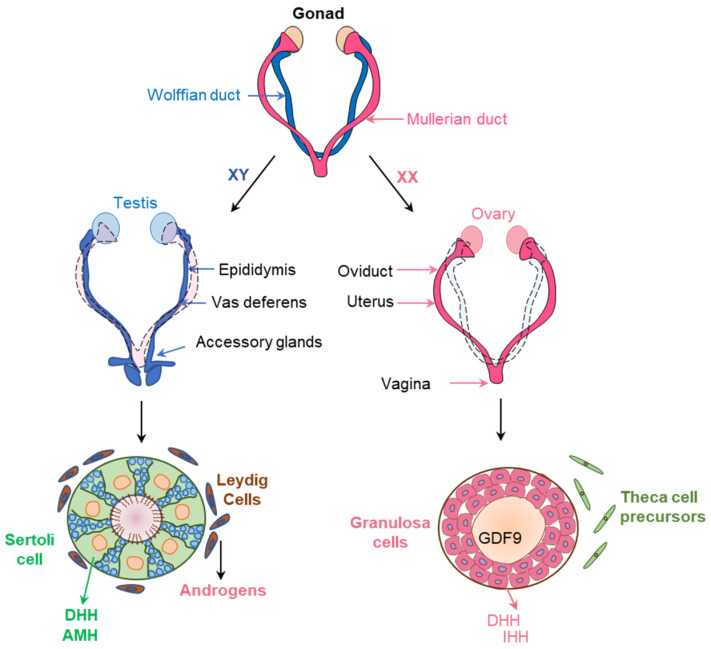
Before sex differentiation, the embryos possess primitive female and male reproductive tracts, known as Müllerian ducts (pink) and Wolffian ducts (blue). Expression of SRY and SOX9 in XY embryos induce the proliferation and differentiation of the Sertoli cells (SCs) in embryonic testes to produce AMH and DHH. DHH signaling recruit the fetal Leydig cells (FLCs) and induce differentiation to synthesize androgens. AMH and androgens retains the Wolffian ducts to form the male reproductive organs. On the other hand, absence, or very low levels or AMH and androgens in the XX embryos facilitate the growth of the Müllerian ducts to form female reproductive organs.

**Figure 8 cells-12-00358-f008:**
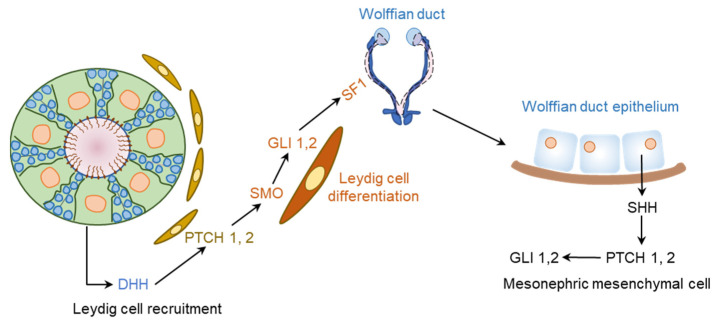
DHH regulation of male gonadal development. The fetal testis is composed of testis cords that contain the developing germ cells and Sertoli cells (SCs), as well as vascular networks. These structures are surrounded by an interstitial tissues compartment containing Leydig cells (LCs). DHH secreted by the SCs signals through PTCH1 and GLI1 in the LCs to regulate their development and overall organization of the testis cord. DHH signaling also induces the expression of SF1 in LCs, which upregulates the expression of steroidogenic enzymes. On the other hand, SHH is expressed in the Wolffian duct epithelium, and the cells in mesonephric mesenchyme express PTCH1, SMO, and GLI1. Thus, the two HH signaling pathways may take part into the epithelial–mesenchymal communication during the male reproductive tract development.

**Figure 9 cells-12-00358-f009:**
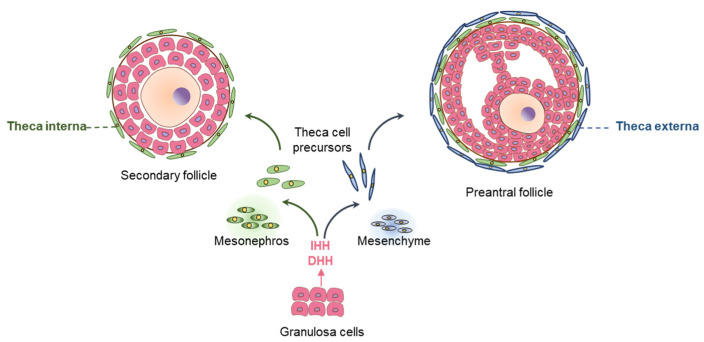
The role of hedgehog signaling in ovarian folliculogenesis. DHH and IHH are expressed in granulosa cells (GCs) of activated ovarian follicles. The major role of HH signaling in ovary involves development and differentiation of theca cells (TCs). Loss of both DHH and IHH in GCs leads to lack of TC development of both mesonephros or mesenchyme origin, associated with ineffective steroidogenesis, and infertility due to failure of ovulation.

**Figure 10 cells-12-00358-f010:**
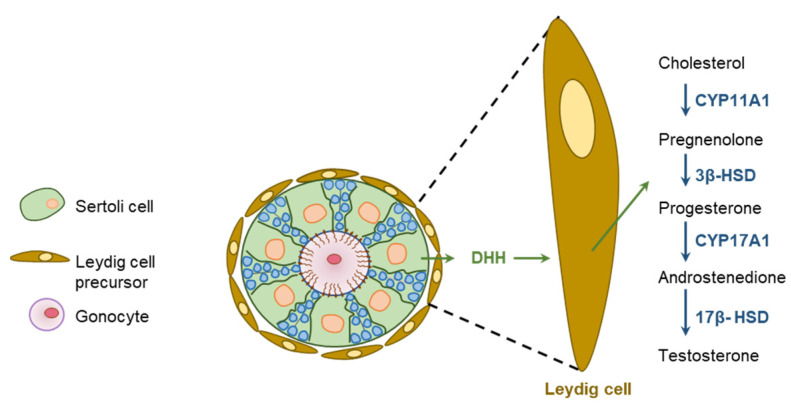
Hedgehog regulation of testicular functions. Sertoli cells (SCs) express DHH and the downstream HH signaling molecules are expressed in the Leydig cells (LCs). DHH secreted from the SCs act of LCs to induce proliferation and differentiation. In response to DHH signaling, LCs initiate steroidogenesis and secrete testosterone.

**Figure 11 cells-12-00358-f011:**
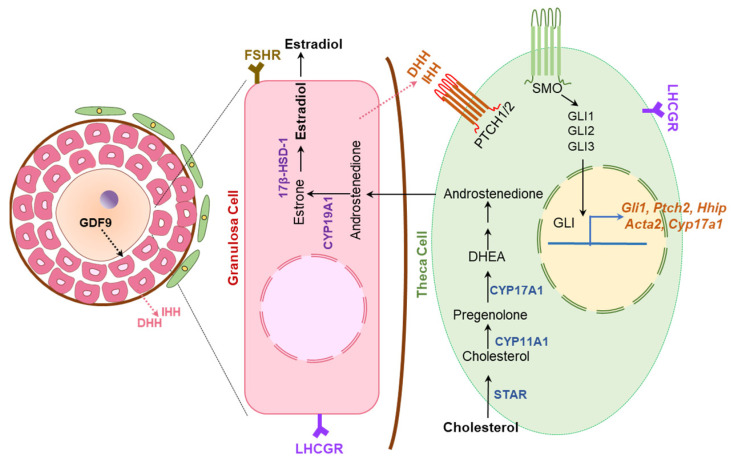
Hedgehog regulation of ovarian steroidogenesis. Granulosa cells (GCs) in activated ovarian follicles express DHH and IHH. The HH molecules secreted from the GCs bind the PTCH receptors expressed on theca cell (TC) precursors to induce their differentiation. Development and differentiation of TCs are essential for steroidogenesis and follicle maturation beyond early antral stage. While GDF9 expressed by oocytes act on GCs to induce DHH and IHH expression, the HH molecules are essential for TC functions. Thus, HH signaling establishes a signaling link among the germ cells and somatic cells of ovarian follicles.

## Data Availability

Not applicable.

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
