# Peer review of "Hedgehog Signaling in Gonadal Development and Function"

_cells, 2023, doi:10.3390/cells12030358_

Round 1
Reviewer 1 Report
The manuscript represents a detailed review of the role of the hegdehog signal in mammalian development, with more detail on the development of sexual dimorphism. The text and figures are clear and with very good detail and information. This review advances knowledge described in previous studies and reviews, which are adequately referenced.
In figure 1 is represented the development of the processes on mouse embryo diagrams. It is necessary to indicate the species to which reference is made in the caption of the corresponding figure, as well as in the text when developmental times are described.
Genes, molecules and structures are described by means of abbreviations and in some paragraphs it is confusing to follow the description, it would facilitate the comprehension to make reference to what type of element is being referred to.
Author Response
Authors response to the reviewer’s comments
We have addressed the reviewers’ concern and revised our manuscript according to their suggestions. The reviewers’ suggestions have substantially improved our manuscript and we hope that the revised manuscript is suitable for publication in Cells.
Response to the reviewer #1
Overall comment: The manuscript represents a detailed review of the role of the hedgehog signal in mammalian development, with more detail on the development of sexual dimorphism. The text and figures are clear and with very good detail and information. This review advances knowledge described in previous studies and reviews, which are adequately referenced.
Response: We appreciate the reviewer’s comments.
Query 1: In figure 1 is represented the development of the processes on mouse embryo diagrams. It is necessary to indicate the species to which reference is made in the caption of the corresponding figure, as well as in the text when developmental times are described.
Response to query 1: We agree with the reviewer. Accordingly, we have corrected the errors in our revised manuscript.
Query 2: Genes, molecules and structures are described by means of abbreviations and in some paragraphs, it is confusing to follow the description, it would facilitate the comprehension to make reference to what type of element is being referred to.
Response to query 2: We appreciate the reviewers suggestion. We have made the necessary corrections in our revised manuscript so that the genes and molecules are easily understandable to the readers.
Reviewer 2 Report
Review of Dilower et al.
Summary
The authors provide an interesting review of the hedgehog (HH) signalling pathway and its functions in gonadal development and in ovary biology. The authors include clear descriptions of the different HH pathways (SHH, DHH, and IHH) and they note their different roles in development. The roles of defective HH signalling in disease is well articulated with many examples and appropriate citations. The review also includes detailed descriptions of post-translational modifications of HH proteins. The schematic figures are expertly illustrated and enhance the quality of the review. Overall, this is an excellent and engaging review of recent developments in HH signaling as it relates to gonadal development. Following some minor revisions, the article will be a wonderful contribution to the journal Cells.
Comments
1. It would help the reader to describe the GLI target genes and downstream pathways regulated by the SHH pathway in more detail.
2. Line 116, introduces the idea that defective SHH signalling can lead to hearing loss in mice. Has this been observed in human patients? Are there mutations in humans that cause similar defects?
Corrections
1. Line 34, spell out GLI in first use.
2. Line 39, roles not role.
3. Grammatical errors, lines 70, 149,
4. Extra spaces should be removed (e.g., lines 84, 88, 138)
5. Sometimes spaces are lacking between the sentence and references (e.g., line 97).
6. Line 228, spell out “E11.5” abbreviation in first use.
Author Response
Authors response to the reviewer’s comments
We have addressed the reviewers’ concern and revised our manuscript according to their suggestions. The reviewers’ suggestions have substantially improved our manuscript and we hope that the revised manuscript is suitable for publication in Cells.
Response to the reviewer #2
Query 1: It would help the reader to describe the GLI target genes and downstream pathways regulated by the SHH pathway in more detail.
Response to query 1: We agree with the reviewer’s suggestion, and we have expanded the section to describe the GLI target genes. We also added to the SHH signaling section. It needs to be indicated that this review article is focused on the hedgehog signaling in the gonads. However, SHH is minimally expressed in the gonads, thus, SHH signaling may not be involved directly in regulating gonadal functions.
Query 2: Line 116, introduces the idea that defective SHH signaling can lead to hearing loss in mice. Has this been observed in human patients? Are there mutations in humans that cause similar defects?
Response to query 2: We could not find any study in support of such pathophysiology. Although suspected to be a likely condition, defective SHH signaling has not yet been found in patients with hearing loss.
Query 3: Corrections
- Line 34, spell out GLI in first use.
- Line 39, roles not role.
- Grammatical errors, lines 70, 149,
- Extra spaces should be removed (e.g., lines 84, 88, 138)
- Sometimes spaces are lacking between the sentence and references (e.g., line 97).
- Line 228, spell out “E11.5” abbreviation in first use.
Response to query 3: We remain thankful for the reviewer’s comments. We have corrected the errors.